# Molecularly Imprinted Polymers Exhibit Low Cytotoxic and Inflammatory Properties in Macrophages *In Vitro*

**Louise Sternbæk** [1,2,3,*], **Martha Kimani** [4], **Kornelia Gawlitza** [4], **Knut Rurack** [4], **Birgit Janicke** [3], **Kersti Alm** [3], **Anette Gjörloff Wingren** [1,2] and **Håkan Eriksson** [1,2]

1  Department of Biomedical Sciences, Faculty of Health and Society, Malmö University, SE-205 06 Malmö, Sweden; anette.gjorloff-wingren@mau.se (A.G.W.); hakan.eriksson@mau.se (H.E.)
2  Biofilms-Research Center for Biointerfaces, Malmö University, SE-205 06 Malmö, Sweden
3  Phase Holographic Imaging AB, SE-223 63 Lund, Sweden; birgitjanicke_1@hotmail.com (B.J.); kersti.alm@phiab.se (K.A.)
4  Chemical and Optical Sensing Division, Bundesanstalt für Materialforschung und-prüfung (BAM), DE-12489 Berlin, Germany; martha-wamaitha.kimani@bam.de (M.K.); kornelia.gawlitza@bam.de (K.G.); knut.rurack@bam.de (K.R.)
*  Correspondence: louise.sternbeak@mau.se

**Abstract:** Molecularly imprinted polymers (MIPs) against sialic acid (SA) have been developed as a detection tool to target cancer cells. Before proceeding to *in vivo* studies, a better knowledge of the overall effects of MIPs on the innate immune system is needed. The aim of this study thus was to exemplarily assess whether SA-MIPs lead to inflammatory and/or cytotoxic responses when administered to phagocytosing cells in the innate immune system. The response of monocytic/macrophage cell lines to two different reference particles, Alhydrogel and PLGA, was compared to their response to SA-MIPs. *In vitro* culture showed a cellular association of SA-MIPs and Alhydrogel, as analyzed by flow cytometry. The reference particle Alhydrogel induced secretion of IL-1β from the monocytic cell line THP-1, whereas almost no secretion was provoked for SA-MIPs. A reduced number of both THP-1 and RAW 264.7 cells were observed after incubation with SA-MIPs and this was not caused by cytotoxicity. Digital holographic cytometry showed that SA-MIP treatment affected cell division, with much fewer cells dividing. Thus, the reduced number of cells after SA-MIP treatment was not linked to SA-MIPs cytotoxicity. In conclusion, SA-MIPs have a low degree of inflammatory properties, are not cytotoxic, and can be applicable for future *in vivo* studies.

**Keywords:** molecularly imprinted polymers; digital holographic cytometry; cytotoxicity; pro-inflammatory cytokines

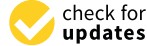



## 1. Introduction

Nanoparticle-assisted cancer detection and monitoring have the potential for wider use in tumor diagnostics and treatment. We and others have previously investigated molecularly imprinted polymers (MIPs) that target different structures, such as glycans, peptides or proteins on the cell surfaces [1–6]. Specifically, the monosaccharide sialic acid (SA) has been the focus of many studies, due to its role as a potential cancer biomarker [7–11]. SA-MIPs have been demonstrated to target different cell lines *in vitro*, as shown by flow cytometry and cell imaging experiments. MIPs have also been frequently used and investigated *in vivo*, with promising results for both targeting and drug delivery [2,12,13]. However, to continue using MIPs *in vivo*, further information regarding the overall effects of MIPs on the innate immune system is needed.

It is well documented that the physicochemical properties of nano- and submicron-sized particles are likely to influence their biological fate and actions of the particles *in vivo* [14]. Nanoparticles often aggregate under physiological conditions, and it is known that aggregates of nanoparticles of an intermediate size (1–5 μm) are phagocytosed more

readily than smaller or larger particles [15]. At physiological conditions, phagocytosis of synthetic particles by monocytes and macrophage-derived cells plays a significant role in determining the fate of delivery systems utilizing nanoparticles [16,17]. Phagocytosis is responsible for the failure of many drug delivery strategies using nanoparticles *in vivo* due to undesirable immune cell targeting [18]. Moreover, protein adsorption to the surface due to the surface charge of the particles affects the interaction with the cells [15]. If particles are removed by macrophages before reaching their intended destination *in vivo* they can induce an inflammatory response due to their interaction with phagocytosing cells. Therefore, understanding and monitoring nanoparticle phagocytosis by immune cells is critical for developing these particles for successful use *in vivo*.

Most leukocytes, including monocytes and macrophages express SA, and thereby SA-MIPs will target SA on those cells. The aim was to study the general impact of MIPs on the innate immune system regarding phagocytosing cells, using SA-MIPs. To study the impact of SA-MIPs on phagocytic cells, the monocytic/macrophage cell lines THP-1 and RAW 264.7 were chosen as classical phagocytosing cells. THP-1 and RAW 264.7 cells have both been extensively used to study macrophage functions, mechanisms, signaling pathways, and drug transport [19,20]. SA-MIPs show the potential to interact with the cells, thus it is relevant to use SA-MIPs together with two different and well-characterized reference particles; aluminum adjuvant in the form of Alhydrogel® [21] and poly(lactic-co-glycolic acid) (PLGA) [22], for interaction studies with phagocytosing cells.

Inflammatory responses, elicited by nanoparticles, in particular, are important to control, since constant or hyperstimulation can result in chronic diseases [14]. Macrophages are central mediators of innate immunity linked to chronic inflammation including anti-oxidative and pro-inflammatory responses as well as cell death [23]. The reference particles Alhydrogel and PLGA were used to compare the possible inflammatory response and cytotoxicity caused by the SA-MIPs. To evaluate any inflammatory response by the SA-MIPs and the reference particles, induction and secretion of the pro-inflammatory cytokines IL-1$\beta$, TNF-$\alpha$ and IL-6 were investigated.

Cell proliferation upon culture in the presence of SA-MIPs was scrutinized using digital holographic cytometry (DHC). DHC is a non-phototoxic quantitative phase imaging technique that enables the monitoring of living cells [24]. The DHC technique allows for long time monitoring of the cells, acquiring high time resolution images, which can be used for longitudinal tracking of individual cells [25,26]. Tracking of individual cells using DHC demonstrated that the SA-MIPs had an impact on cell proliferation and cell division, but the reduction in cell numbers could not be attributed to SA-MIPs cytotoxicity.

In this study, we show that the induced release of inflammatory cytokines with SA-MIPs was low in comparison with the reference particles Alhydrogel and PLGA. Moreover, the SA-MIPs were not cytotoxic to the cells, but the cell cycle was affected by the presence of the SA-MIPs as determined using DHC.

## 2. Materials and Methods

### 2.1. Preparation of SA-MIPs

The hybrid polymer probes (SA-MIPs) were prepared according to a previously described protocol by Shinde et al. [4] but using deprotonated SA as the template to improve the binding properties of the MIPs. Deprotonated SA was prepared by dissolving equimolar amounts of SA and tetrabutylammonium hydroxide (Sigma-Aldrich, St. Louis, MO, USA, ≥99.0%, CAS 147741-30-8) in acetonitrile. After 30 min of mixing, the mixture was placed in a vacuum concentrator and dried overnight to give the corresponding template. The few-nanometer thin SA-MIP shells grafted on submicron-sized silica cores were equipped with nitrobenzoxadiazole (NBD) fluorescent reporter groups allowing environmentally sensitive fluorescence detection at 530 ± 15 nm. Dried SA-MIPs were resuspended in phosphate saline buffer (PBS, Invitrogen, Waltham, MA, USA), to a stock solution of 1.0 mg/mL. The thickness of the polymer shells and the MIP's binding properties were similar to the ones described in Shinde et al. [4].

## 2.2. Reference Particles

The aluminum adjuvant preparation used herein was AlO(OH)-based Alhydrogel [®], purchased from Brenntag Biosector (Frederikssund, Denmark). Lumogallion was obtained from TCI Europe N.V. (Antwerp, Belgium (CAS 4386-25-8)) and Alhydrogel was labeled with lumogallion according to Mile et al. [27], prepared prior to this study.

PLGA microspheres of 2 μm diameter were obtained from (Sigma-Aldrich, St. Louis, MO, USA).

## 2.3. Cell Culturing

RAW 264.7 cells (ATCC[®] TIB-71TM, ATCC LGC Standards, Teddington, UK) were cultured in RMPI-1640 medium (GibcoTM, ThermoFisher Scientific, Grand Island, NE, USA) with addition of 10% fetal calf serum (FBS, Invitrogen, San Diego, CA, USA) and 1% penicillin–streptomycin (Invitrogen). THP-1 cells (ATCC TIP-202, ATCC LGC Standards, Teddington, UK) were cultured in RPMI 1640 medium supplemented with 10% FBS and 100 μg/mL gentamicin (Corning Media Tech, ThermoFisher Scientific, Waltham, MA, USA). This medium will be referred to as R10. All cells were cultured at 37 °C in a humidified atmosphere with 5% $CO_2$. Cells were maintained by sub-culturing once every third day.

## 2.4. Size Distribution of SA-MIPs and Alhydrogel

SA-MIPs and Alhydrogel were incubated overnight in R10 culture medium at a concentration of 500 μg/mL. The size distribution of the particles was determined using a Coulter LS230 Laser Diffraction Particle Size Analyzer equipped with a Small Volume Module (Beckman Coulter, Brea, CA, USA).

## 2.5. Protein Adsorption

SA-MIPs and Alhydrogel were incubated in 3 mL R10 overnight at 400 μg/mL and 1 μg/mL in a 12-well plate. Each sample was resuspended, and 1 mL was withdrawn and harvested by centrifugation for 5 min at $13,000 \times g$. The pellets were washed by resuspension in 1 mL PBS and collected by centrifugation for 5 min at $13,000 \times g$. Finally, the pellets were resuspended in 100 μL Laemmli protein sample buffer containing dithiothreitol (DTT) (Bio-Rad Laboratories, Hercules, CA, USA). The proteins forming the corona on the SA-MIPs and Alhydrogel were identified by SDS-PAGE using NuPAGE 4–12% Bis-Tris gel (1.0 mm × 15 wells, Invitrogen, ThermoFisher Scientific, Waltham, MA, USA) and SDS-Running Buffer (NuPAGE MES SDS Running Buffer, Invitrogen, ThermoFisher Scientific). Samples and molecular weight marker (See Blue Pre-Stained Protein Standard, Invitrogen, ThermoFisher Scientific) were run at a constant voltage of 180 V. Protein bands were visualized by Coomassie Brilliant Blue R-250 staining solution (Bio-Rad Laboratories, Hercules, CA, USA).

## 2.6. Flow Cytometry Analysis
THP-1 Cells

Triplicates of THP-1 100 μL cells $0.5 \times 10^6$ cells/mL were co-cultured in a 96-well plate with 0 μg/mL, 25 μg/mL, 50 μg/mL or 100 μg/mL SA-MIPs, Alhydrogel or PLGA. The cells were incubated at 37 °C with 5% $CO_2$ for 24 h. After incubation, the triplicated samples were harvested and pooled with 600 μL 0.1% BSA in PBS before the cells were counted by flow cytometry (Accuri C6 Flow Cytometer, BD Bioscience, Franklin Lakes, NJ, USA). Each sample was then centrifuged at $300 \times g$ for 5 min and resuspended in 300 μL 1% paraformaldehyde (PFA) in PBS and the fluorescence intensity was analyzed by flow cytometry (Accuri C6 Flow Cytometry, BD Bioscience).

## 2.7. RAW 264.7 Cells

Samples with $1 \times 10^6$ RAW 264.7 cells/mL were seeded in a 24-well plate and incubated overnight at 37 °C with 5% $CO_2$, for cells to adhere. After incubation, the cells were incubated with either 0 μg/mL, 25 μg/mL, 50 μg/mL or 100 μg/mL SA-MIPs or Alhydro-

gel. The cells were incubated at 37 °C with 5% $CO_2$. RAW 264.7 cells were harvested by trypsinization after 24 h. The cell suspension was washed three times with 2 mL PBS at $300\times g$ for 5 min, and the cells were resuspended in 300 μL PBS and analyzed using flow cytometry (Accuri C6 Flow Cytometry, BD Bioscience, Franklin Lakes, NJ, USA).

### 2.8. Fluorescence Microscopy

RAW 264.7 cells were seeded into culture chambers (BD Falcon CultureSlides, uncoated from BD Bioscience, Franklin Lakes, NJ, USA) using 200 μL $0.5 \times 10^6$ cells/mL in each chamber and the cells were allowed to adhere overnight at 37 °C with 5% $CO_2$. Next day, the cells were washed with 200 μL culture medium and 200 μL 100 μg/mL SA-MIPs in cell culture medium was added to the chamber, or the cells were left unstained as a control and incubated overnight at 37 °C with 5% $CO_2$. Each chamber was washed three times with 300 μL PBS and fixed with 100 μL 4% PFA for 10 min at RT. The cells were washed once with PBS and once with 0.05% TritonX-100 (Sigma-Aldrich, St. Louis, MO, USA) in PBS, after which the cells were permeabilized with 0.05% TritonX-100 in PBS for 10 min. The buffer was removed and replaced with 100 μL of rhodamine-phalloidin red (Sigma-Aldrich) diluted 1:100 in PBS. After 30 min at RT in the dark, they were washed once with 0.05% TritonX-100 in PBS and twice with PBS. Finally, the cells were mounted using ProLong® Gold Antifade Mounting with DAPI (Life Technologies, ThermoFisher Scientific, Waltham, MA, USA) and analyzed by fluorescence microscopy (AX70 Olympus, Littleton, MA, USA).

### 2.9. Stimulation of THP-1 Cells

Firstly, 5 mL of THP-1 cells, $0.8 \times 10^6$ cells/mL were pre-incubated in two T25 flasks: one with 1 μg lipopolysaccharide (LPS) per ml in R10 and one with R10 medium at 37 °C with 5% $CO_2$ for 4 h. The cells were then harvested by centrifugation, resuspended in R10 and triplicates of 100 μL $1 \times 10^6$ cells/mL were cultured with an equal volume of 50 μg/mL, 100 μg/mL or 200 μg/mL SA-MIP, Alhydrogel or PLGA overnight at 37 °C with 5% $CO_2$. The triplicates of the different stimulations were collected and centrifuged for 5 min at $300\times g$, whereafter the supernatants were collected and centrifuged for 10 min at $13,000\times g$. The final supernatants were collected and stored at $-20$ °C for later use.

### 2.10. Stimulation of RAW Cells

RAW 264.7 cells, $1 \times 10^4$ per well, were seeded in triplicates into a 96-well plate and stimulated with 25 μg/mL, 50 μg/mL or 100 μg/mL SA-MIP, Alhydrogel or PLGA for 24 h at 37 °C with 5% $CO_2$. The medium from triplicates of the stimulated cells was pooled and centrifuged at $300\times g$ for 10 min. The supernatants were stored at $-20$ °C until later use.

### 2.11. Detection of IL-1β, IL-6 and TNF-α by Enzyme-Linked Immunosorbent Assay (ELISA)

DuoSet sandwich ELISAs were used to measure human IL-1β, mouse IL-6 and mouse tumor necrosis factor alpha (TNF-α). All reagents and instructions were supplied by Bio-Techne (R&D Systems, Minneapolis, MN, USA). The supernatants of THP-1 and RAW 264.7 cells were thawed, and the cytokine secretion was measured according to the manufacturer's instructions. Each plate was assayed by SpectraMax iD5 (Molecular Devices, San Jose, CA, USA) and data were analyzed by SoftMax software (vPro 7.0,3 Molecular Devices, San Jose, CA, USA).

### 2.12. Lactate Dehydrogenase (LDH) Activity

THP-1 and RAW 264.7 cells, $1 \times 10^4$ per well, were incubated with 0 μg/mL, 25 μg/mL, 50 μg/mL or 100 μg/mL SA-MIP, Alhydrogel or PLGA particles for 24 h before the LDH activity was determined in the medium, according to the manufacturer's instructions (Invitrogen, MA, USA). Briefly, 50 μL of each sample was incubated with 50 μL of the kit reaction mixture for 30 min at RT in the dark. Stop solution, 50 μL, was added to each sample and the absorbance of the samples was then measured at 490 nm and 680 nm in a plate reader (SpectraMax iD5 Molecular Devices, San Jose, CA, USA).

*2.13. Digital Holographic Cytometry (DHC) and Cell Tracking*

RAW 264.7 cells were seeded in a Sarstedt lumox® 96-well plate (Sarstedt, Germany) with $2 \times 10^4$ cells per well and incubated overnight at 37 °C with 5% $CO_2$ allowing the cells to adhere. The medium in each well was renewed with either 40 μg/mL SA-MIPs, for treated cells, or culture medium for untreated cells. The lid was changed to Hololids for 96-well plates (PHIAB, Lund, Sweden), which enable cell imaging by the HoloMonitor M4 digital holographic cytometer (DHC) (PHIAB). The plates were placed on the motorized stages of HoloMonitor M4 in a standard 37 °C with 5% $CO_2$ incubator. Triplicated experiments were performed using 2 wells of treated and untreated cells in each experiment. Five positions in each well were chosen and images captured every 15 min for 48 h. Time-lapse imaging, image processing, segmentation and analysis were conducted with the App Suite software package (v3.5.0, PHIAB, Lund, Sweden) based on absolute values of refractive indexes for cells and culture medium. As the laser intensity is approximately 10 W/m$^2$ during imaging, and exposure time is less than 10 ms, it is assumed that the laser irradiation has only minimal effect on the physiological functions of the cells. App Suite was used for the tracking of individual cells. For each capturing position, cells were selected at time-point 0 h, which is the first image acquired in the 48 h time-lapse. When the cells divided, the two daughter cells were selected for further analysis and connected to the mother cell. This enables the calculation of the time of each cell division for individual cells, [25,26]. Based on known time-points for cell divisions and from which cell the daughter cells originated, a cell family tree was drawn for each cell selected at time-point 0 h using RStudio software (v.2021.09.1, RStudio, Boston, MA, USA).

*2.14. Statistical Methods*

Mean and standard deviation (SD) were used for statistical analysis of all calculations. Mann–Whitney rank-sum test using GraphPad Prism 9 (San Diego, CA, USA) was used. The computer language RStudio was used for drawing cell family trees.

## 3. Results

MIPs have been used extensively *in vitro* for different biological applications [6] and have the potential to be used *in vivo*. However, prior to *in vivo* use, the impact of SA-MIPs on phagocytosing cells must be evaluated. To assess the influence of SA-MIPs on phagocytic cell lines, the monocytic/macrophage cell lines THP-1 and RAW 264.7 were chosen as classical examples of phagocytosing cells.

*3.1. Particle Characteristics*

Particles of an aluminum adjuvant labeled with lumogallion (Alhydrogel) and particles of PLGA were included in the study as reference particles, both model particles with well-documented use both *in vivo* and *in vitro*. Some general properties of SA-MIPs and the two reference particles are summarized in Table 1.

**Table 1.** Particle characterization of sialic acid molecularly imprinted polymers (SA-MIPs), Alhydrogel and poly(lactic-co-glycolic acid) (PLGA).

|  | SA-MIP | Alhydrogel | PLGA |
|---|---|---|---|
| **Overall surface charge** | Negative [28,29] | Positive [30] | Negative [14] |
| **Single particle diameter** | ~0.2 μm [4] | ~0.05 μm [31] | ~2 μm [22] |
| **Particle size in medium (form aggregates)** | 0.2–3 μm * | 0.5–4 μm * [30] | No aggregation ** |
| **Protein adsorption in medium** | ~50 kDa *** | ~ 50 kDa–198 kDa *** Several bands | Not investigated |
| **Fluorescence dye** | Nitrobenzoxadiazole [4] | Lumogallion [27] | None |

* Determined by Coulter LS230 Laser Diffraction Particle Size Analyzer equipped with a Small Volume Module (Beckman Coulter, CA, USA) ** determined by flow cytometry *** determined by SDS-PAGE.

The prepared SA-MIPs form micron-sized aggregates when the stock solution is diluted in the culture medium and are expected to be negatively charged at pH 7 [28,29]. A similar aggregation of nanoparticles to micron-sized features is observed for the positively charged reference particle Alhydrogel. PLGA is a negatively charged microparticle that does not aggregate as seen by flow cytometry. The size of the unaggregated PLGA particles is close to the size of aggregated SA-MIPs and Alhydrogel particles.

### 3.2. Association of SA-MIPs and Phagocytosing Cells

Flow cytometry revealed an unambiguous cellular association between monocytic/macr ophage cells and SA-MIPs after 24 h of incubation. The association is displayed in flow cytometry histograms showing the association of the NBD fluorophore-containing SA-MIPs (red histogram) or the lumogallion fluorophore-labeled Alhydrogel (blue histogram) with each cell line (Figure 1). The distribution of cells according to their fluorescence is seen on the x-axes with the black histograms in Figure 1 showing the background/autofluorescence of the cells in the absence of particles). Upon association with the fluorescent particles, the cellular fluorescence will increase, shifting the histogram to higher FL1/FL2 values, and the percentage of cells associated with particles can be estimated. The results show that 100% of the cells displayed an increased fluorescence upon incubation with Alhydrogel and hence are associated with the Alhydrogel particles (blue histogram Figure 1), whereas only 20–30% of the cells became associated with the SA-MIPs (red histogram Figure 1).

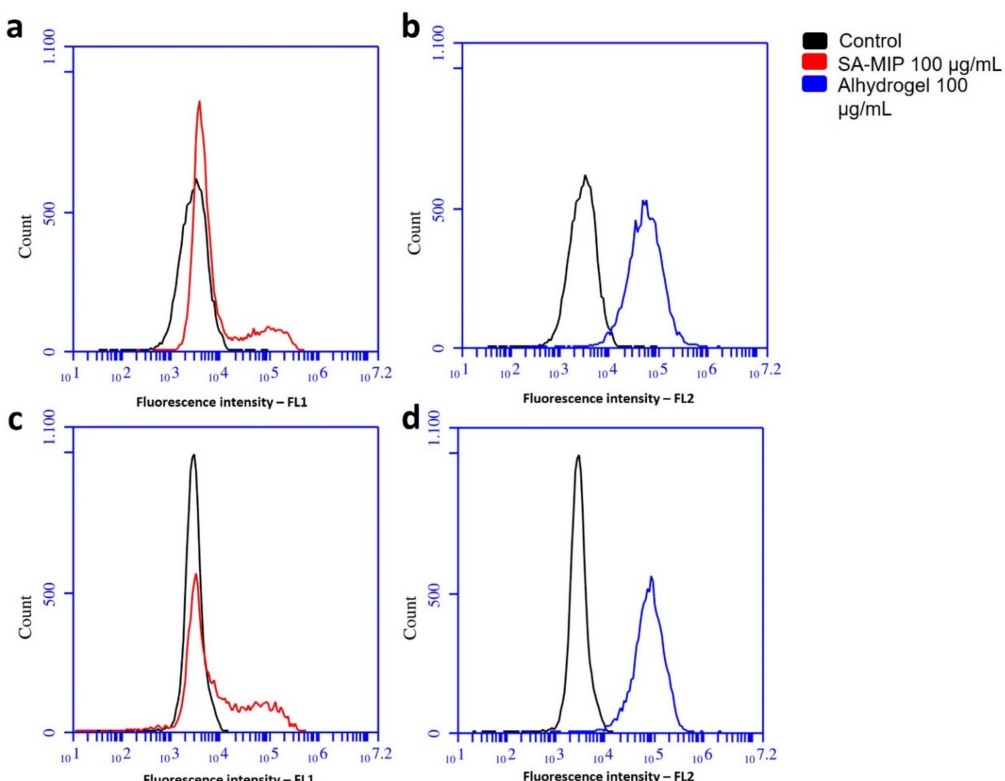

**Figure 1.** SA-MIPs had a lower association with phagocytosing cells compared to Alhydrogel. THP-1 (**a**,**b**) and RAW 264.7 (**c**,**d**) cells cultured for 24 h in medium (control = black lines), or in the presence of 100 µg/mL SA-MIPs (**a**,**c**; red lines) or 100 µg/mL Alhydrogel (**b**,**d**; blue lines). The histograms show the fluorescence intensity. SA-MIPs (NBD) are measured in the FL-1 channel (530 ± 15 nm) and Alhydrogel (lumogallion) was measured in the FL-2 channel (585 ± 20 nm). One representative experiment out of at least three independent experiments is shown.

Fluorescence microscopy was used to visualize a potential intracellular location of the SA-MIP particles. Membrane staining of RAW 264.7 cells using actin filament staining with rhodamine-phalloidin, SA-MIPs, and nuclear staining using DAPI is shown in Figure 2.

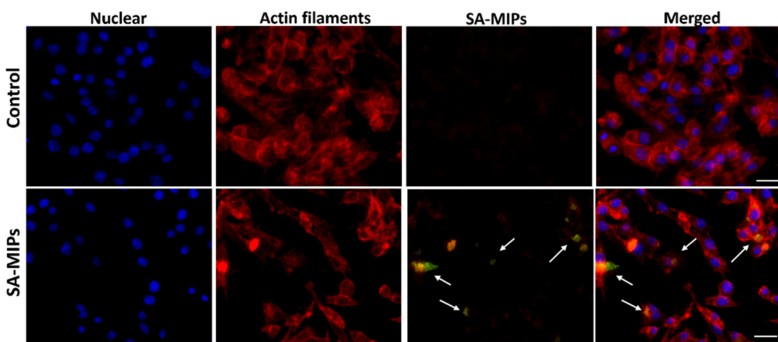

**Figure 2.** Fluorescence microscopy images of the SA-MIPs' potential intracellular location in RAW 264.7 cells. Prior to fixation, the cells were incubated for 24 h with culture medium (control) or 100 μg/mL SA-MIPs. Cells were stained for nuclei (DAPI, blue), actin filaments (rhodamine-phalloidin, red) and SA-MIPs (NBD, green). Scale bar at 10 μm. Arrows mark binding or internalization of SA-MIP. One representative experiment out of at least three independent experiments is shown.

### 3.3. Incubation with SA-MIPs Did Not Result in Increased Cytokine Secretion

THP-1 cells with functional inflammasomes were achieved by pre-stimulation with 1 μg/mL LPS. After 4 h, cells were left with the medium alone or subsequently incubated with three different concentrations of SA-MIPs, Alhydrogel or PLGA, respectively, for 24 h. As a control, cells not pre-stimulated with LPS were incubated with the corresponding concentrations of the particles. After incubation with SA-MIPs or PLGA, limited secretion of IL-1β could be determined, whereas Alhydrogel induced high levels of secreted IL-1β from the THP-1 cells (Figure 3, left). Without pre-incubation with LPS, no activation of the inflammasomes took place (Figure 3, right) resulting in the absence of secreted IL-1β. SA-MIPs induced low levels of secreted TNF-α and IL-6 in the mouse RAW 264.7 cells. The levels of excreted TNF-α and IL-6 were in the same range as for the two reference particles Alhydrogel and PLGA (Figure 4).

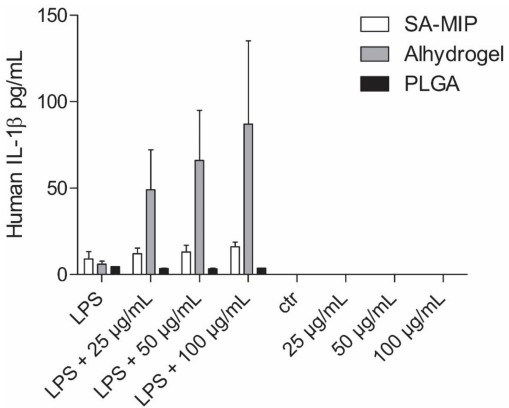

**Figure 3.** Secretion of Il-1β after incubation with SA-MIPs, Alhydrogel and PLGA. THP-1 cells were either primed with LPS 1 μg/mL for 4 h at 37 °C or incubated with cell culture medium alone before the cells were harvested and incubated with 0 μg/mL, 25 μg/mL, 50 μg/mL or 100 μg/mL SA-MIPs, Alhydrogel or PLGA. Media from the incubations were collected after 24 h and the content of IL-1β was analyzed using ELISA. The left part of the graph shows THP-1 cells primed with LPS, and the right part is the unprimed control. The results represent the mean of three independent experiments ±SD.

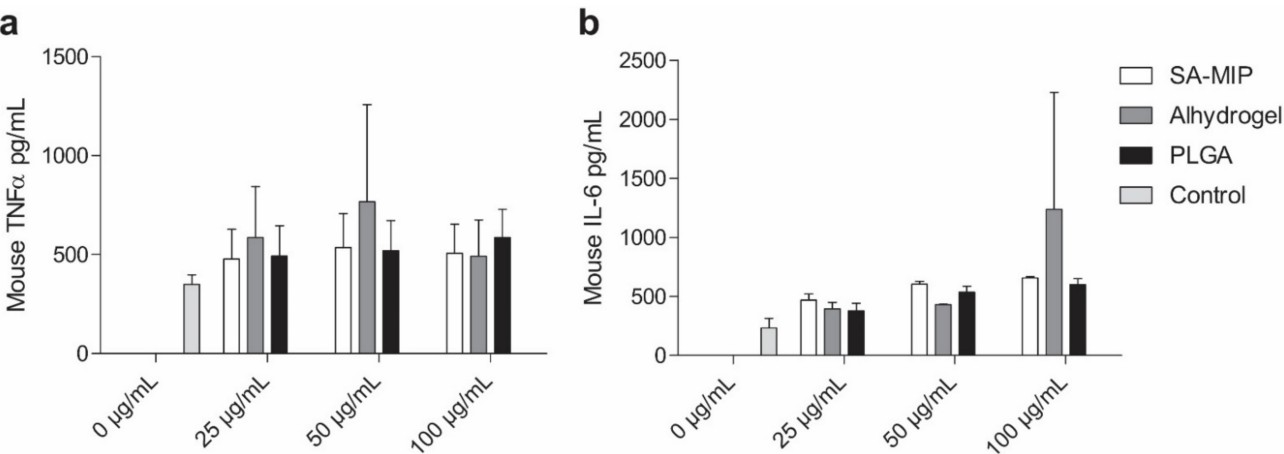

**Figure 4.** Secretion of TNF-α and IL-6 after incubation with SA-MIPs, Alhydrogel and PLGA. RAW 264.7 cells were cultured in medium alone or with additions of 0 µg/mL, 25 µg/mL, 50 µg/mL or 100 µg/mL SA-MIPs, Alhydrogel or PLGA, respectively, for 24 h at 37 °C. Culture supernatants were assayed for TNF-α (**a**) and IL-6 (**b**) cytokines by ELISA. The results represent the mean of three independent experiments ±SD.

### 3.4. SA-MIPs Cause Decreased Cell Proliferation, but Are Not Cytotoxic

For cell proliferation studies, THP-1 cells were incubated with three different concentrations of SA-MIPs, Alhydrogel or PLGA, respectively, for 24 h. A reduced number of THP-1 cells was observed upon incubation with the SA-MIPs and the reference particles (Figure 5a). Since the growth attenuation could indicate particle cytotoxicity, the LDH activity in the culture medium was analyzed. The results show that none of the particles were cytotoxic to THP-1 cells; although, a very slight increase in LDH activity was observed upon addition of the highest concentration of Alhydrogel, 100 µg/mL (Figure 5b).

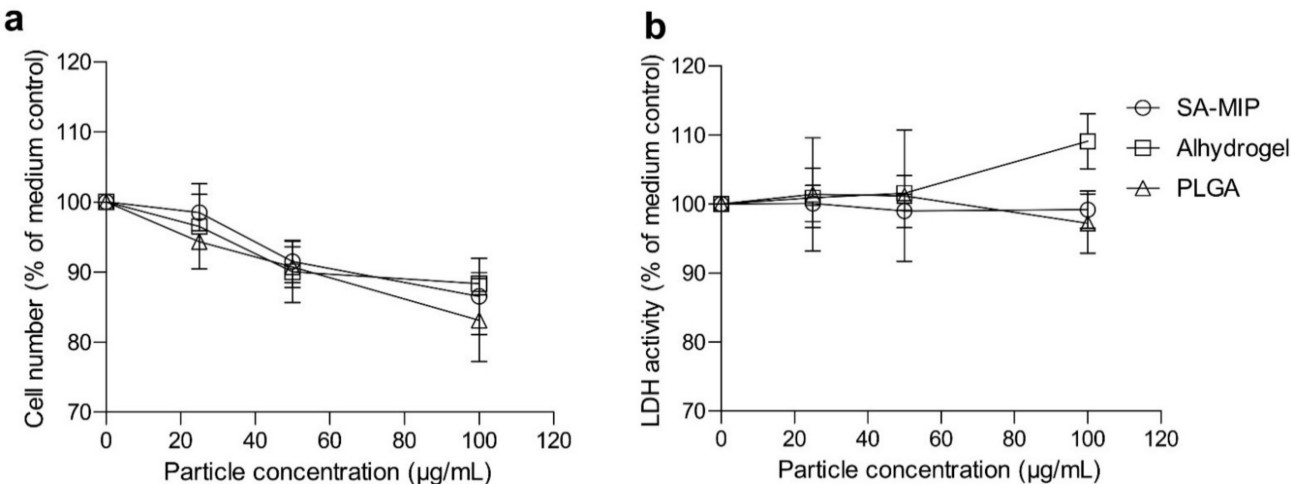

**Figure 5.** Viable THP-1 cells after incubation with SA-MIPs, Alhydrogel or PLGA particles for 24 h. THP-1 cells were either counted (**a**) or the release of lactate dehydrogenase (LDH) into the culture medium was measured (**b**). THP-1 cells were incubated with 0 µg/mL, 25 µg/mL, 50 µg/mL or 100 µg/mL SA-MIPs, Alhydrogel or PLGA for 24 h. Results shown as % of the medium control and the average of three independent experiments with ±SD.

Real-time holographic images were captured every 15 min during 48 h of RAW 264.7 cells cultured in the presence or absence of SA-MIPs. Examples of holographic images of cells incubated with 0 µg/mL SA-MIPs and 40 µg/mL SA-MIPs from a randomly selected area in the culture well at the first and last time-point of the time-lapse are shown

(Figure 6a, b). SA-MIPs (40 µg/mL) were cultured together with RAW 264.7 cells for up to 48 h and the cell number was determined using DHC (Figure 6c). More than twice as many cells were identified in samples with cells grown in a culture medium only after 48 h compared to cells cultured in the presence of SA-MIPs.

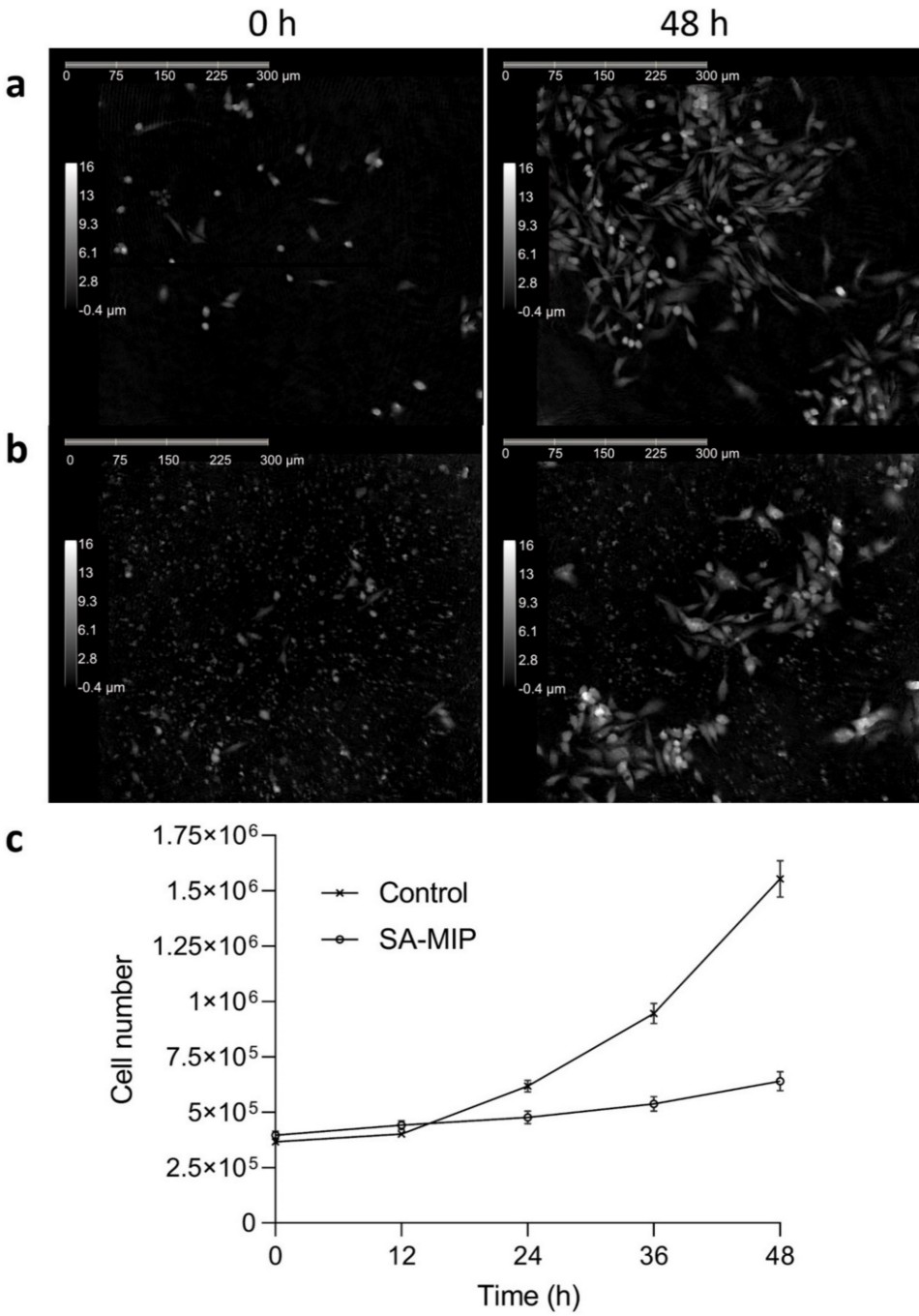

**Figure 6.** DHC images in 2D and cell proliferation. Representative images from DHC time-lapse at time-point 0 h and 48 h of RAW 264.7 cells incubated with medium (**a**) or 40 µg/mL SA-MIPs (**b**). The SA-MIPs were added immediately before the start of the time-lapse and are seen as small dots surrounding the cells. The horizontal scale bars at the top in the images represent 300 µm and the vertical scale bars to the left represent the optical thickness −0.4 to 16 µm. Average number of cells visible in time-lapse frames over 48 h (**c**) of three independent experiments with ±SD. RAW 264.7 cells were seeded with medium for 24 h. Time-lapse imaging was initiated when 40 µg/mL SA-MIPs was added 24 h after cell seeding. This corresponds to time 0 in (**c**).

### 3.5. SA-MIPs Affect Cell Cycle Progression

To further study the effect on proliferation seen at the cell population level, individual longitudinal tracking of RAW 264.7 cells was performed. Cells were cultured in the presence or absence of SA-MIPs, and individual cells were tracked for 48 h. Cells in the first image of every capture position, timepoint 0 h, were identified visually (in total, 109 control cell families and 91 cell families cultured with SA-MIPs) and tracked until the next division. Supplementary Figure S1 shows an example of cell division and segmentation from App Suite. The resulting daughter cells of each cell division were identified and tracking continued, which made it possible to construct cell family trees and to calculate the time between cell divisions for individual cells. A schematic representation of different family trees is shown for RAW 264.7 cells grown in 0 μg/mL SA-MIPs (control, Figure 7a) and 40 μg/mL SA-MIPs (Figure 7b). The complete set of cell family trees can be found in Figure S2. Notable is that the number of cell divisions during the 48 h time-lapse differed between individual cells with the same treatment, and it was also apparent that the cell cycle time varied (Figure 8).

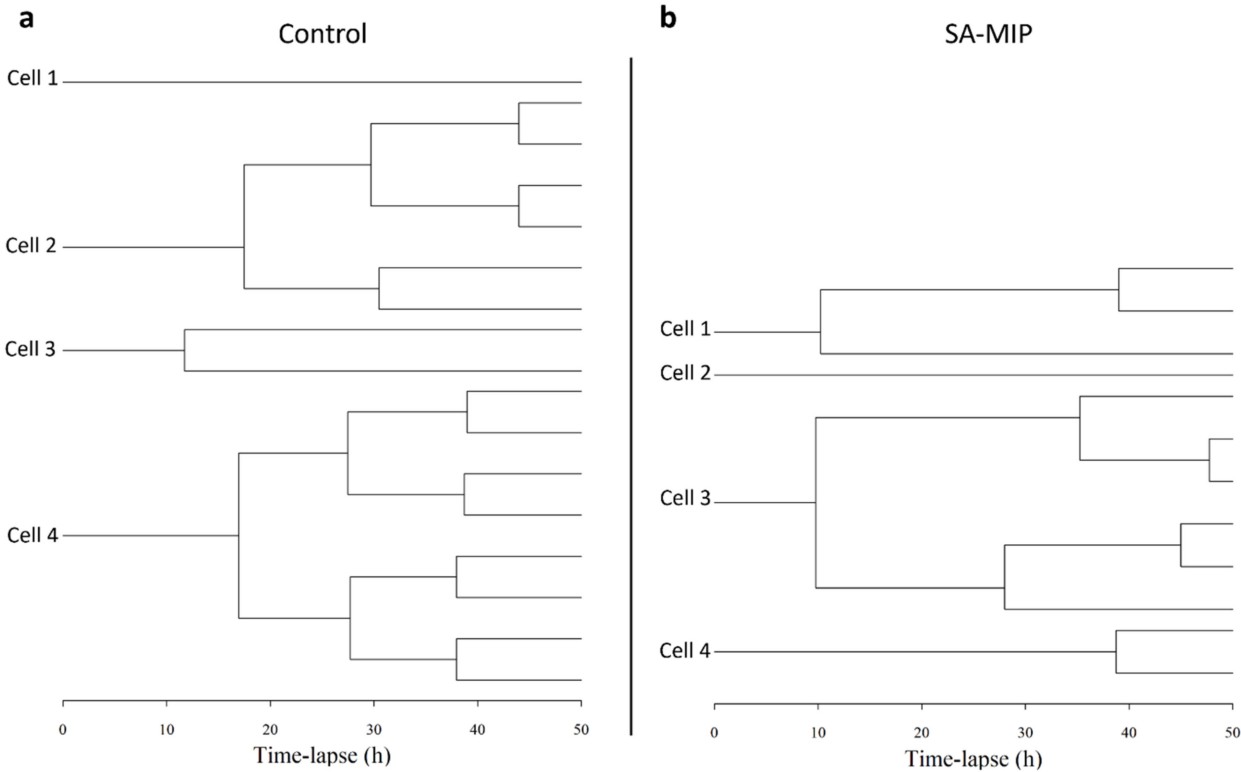

**Figure 7.** Schematic cell family trees for RAW-264.7 cells. RAW 264.7 cells were incubated with culture medium only (**a**) or with 40 μg/mL SA-MIP (**b**) for 48 h. Holographic images were captured every 15 min at several positions, thus creating 48 h time-lapse movies. For every capture position, cells in the first image (0 h) were selected and tracked until division. In the schematic cell family trees, each forking of the line indicates a cell division. The resulting daughter cells of each cell division were identified, and tracking continued throughout the 48 h time-lapse. In total 109 control cell family trees and 91 SA-MIPs treated cell family trees were analyzed.

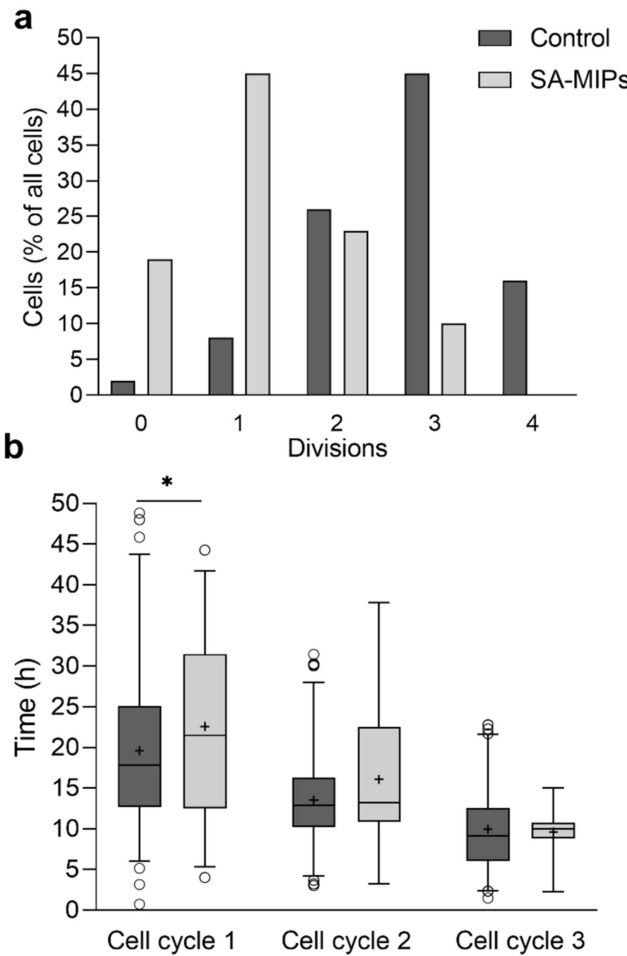

**Figure 8.** Number of cell divisions and length of individual cell divisions in the presence or absence of SA-MIP. Cells were grown with 0 or 40 μg/mL SA-MIPs for 48 h. Images were captured using DHC time-lapse. Cell families were tracked longitudinally using App Suite. Data obtained from longitudinal tracking of cells are presented in supplementary data Figure S1. (**a**) shows the relative numbers of cell families undergoing none, one, two, three or four divisions observed during 48 h for all tracked cells in the first timepoint of the experiment are presented. (**b**) shows the length (h) of individual cell cycles for RAW 264.7 cells with 0 or 40 μg/mL SA-MIPs. In total 109 untreated cell families and 91 SA-MIPs-treated cell families were longitudinally tracked. In the box plot, + marks the mean value, and the whiskers represent the smallest and largest observations (2.5–97.5%) with the circles (O) representing the outliers.

SA-MIPs treatment decreased the number of dividing cells in the cell culture as well as causing cells to divide fewer times. For RAW 264.7 cells cultured in the presence of SA-MIPs, only 10% of the tracked cell families managed to divide three times, while 20% of the cells never divided, and 45% divided only once. For control cells, three divisions were seen for 46% of the control cell families and even four divisions for 17% of the control cell families (Figure 8a). Only 2% of the cells never divided. The mean time between cell divisions was $22.6 \pm 11.8$ h, $16.1 \pm 8.5$ h and $9.6 \pm 3.4$ h, for the respective first, second and third cell cycles of cells cultured in the presence of SA-MIPs (Figure 8b). For cells cultured in a culture medium, the mean time between divisions was $14.3 \pm 8.9$ h, $12.6 \pm 5.5$ h and $22.3 \pm 5.0$ h, respectively. The difference in cell cycle time was only significant for cell cycle 1 with $p = 0.02$, while no significant difference was observed for either cell cycle 2 or 3. This indicates that although only a minor number of cells were dividing upon incubation with SA-MIPs, these dividing cells show the same cell division time as the cells incubated with a culture medium (Figure 8b).

## 4. Discussion

In this study, we have used two phagocytosing cell lines that give a comprehensive indication of the interaction between particles and immune cells of the native immune system. THP-1 and RAW 264.7 cells have both been extensively used to study macrophage functions, mechanisms, signaling pathways and drug transport [19,20]. THP-1 cells are a human monocytic suspension cell line, whereas RAW 264.7 cells are an adherent mouse macrophage cell line. Adherent cells can be assumed to be more efficient in their phagocytosing capability due to their capability of moving and adhering to sedimented particles before phagocytosis [32].

Once nanoparticles enter a physiological environment, the surface of nanoparticles is rapidly covered by proteins to form a corona. Adsorption of proteins to the surface depends on the particle's surface charge, and high amounts of adsorbed protein presuppose an increased cell interaction due to opsonins such as IgG and complement factors [32,33]. A positive surface charge, as for Alhydrogel at pH 7, increases protein adsorption at pH 7, probably increasing the association between cells and particles. The opposite applies to negatively charged particles such as the SA-MIPs [14], which exhibit low protein adsorption after incubation in a serum-containing culture medium (Figure S3). This could explain the difference between SA-MIPs and Alhydrogel, as observed by flow cytometry (Figure 1). While all the cells became associated with Alhydrogel upon incubation with cells, only 20–30% of the cells became associated with SA-MIPs upon incubation. The SA-MIPs have a specificity for SA expressed by the target cells and the present results correspond well with previously reported SA-MIP binding results based on the expression of SA by various cancer cell lines [4,6,7,14,34]. In our study, the focus was not on the binding efficiency of SA-MIPs to its targets, but on the cellular uptake and possible interference of the SA-MIPs with phagocytosing cells.

The association between cells and particles is a pre-requisite for phagocytosis and it is well-documented that the reference particle Alhydrogel is phagocytosed by viable cells [35]. In accordance with flow cytometry, the fluorescence microscopy images showed a limited number of cells associated with the SA-MIPs; although, the images also indicated that the associations of SA-MIPs were both on the cell surface and intracellularly (Figure 2), implying modest phagocytosis of the SA-MIPs by the macrophages; although, the particles had a specificity for the phagocytosing cells.

Cytokines are synthesized and released in response to activation of pattern recognition receptors or by activation of the inflammasomes, something which can occur upon phagocytosis of particles [17]. Although a restricted number of inflammatory cytokines were investigated in this study, the investigated cytokines, IL-1β, IL-6 and TNF-α are major inflammatory cytokines and are expected to be secreted when inflammation is triggered. THP-1 cells do not produce the pro-form of IL-1β and formation of the NALP3 (NLR family pyrin domain containing three) inflammasome unless pre-stimulated with LPS [36,37]. Upon activation of the inflammasomes, caspase 1 is activated, which cleaves the pro-form of IL-1β into IL-1β, which becomes secreted into the medium [38,39]. After pre-stimulation with LPS, THP-1 cells were induced to assemble the NALP3 inflammasome and to synthesize pro-IL-1β. THP-1 cells pre-stimulated with LPS and cultured with SA-MIPs only showed a minor secretion of IL-1β, which means that the inflammasomes and thereby the caspase 1 was not activated by the SA-MIPs. Al-adjuvants are known inducers of the inflammasomes [40–42], and as expected, a high IL-1β secretion was observed in the presence of Alhydrogel.

Several other inflammatory cytokines are also induced and secreted by monocytes and macrophages through activation of the NF-kappaB signaling pathway [33,43]. Future studies of additional cytokines and transcriptional signaling pathways such as NF-kappaB will reveal if these pathways are affected. No distinct dose dependence of secreted TNF-α and IL-6 as a function of particle association was observed for either of the particles. In addition, SA-MIPs treatment did not cause an enhanced cytokine secretion compared to the reference particles, suggesting a limited NF-kappaB mediated inflammatory response

was induced for the SA-MIPs. These results imply that the SA-MIPs possess low-level inflammatory properties that are of the same magnitude as PLGA particles, commonly used as an *in vivo* reference.

Dying cells induce an inflammatory response due to the release of damage-associated molecular patterns (DAMPs) and thereby engagement with pathogen recognition receptors (PRRs) on viable cells [44–46]. A reduced number of cells was observed upon the counting of the cells after culturing in the presence of SA-MIPs (Figures 5a and 6) indicating a possible cytotoxic effect by the SA-MIPs. However, a reduction in the cell number was also observed upon culturing with the reference particles Alhydrogel and PLGA (Figure 5a). PLGA particles are principally regarded as low-toxicity particles used in applications *in vivo* [47,48]. It was only in the presence of the highest concentration of Alhydrogel that a tendency of an increased release of LDH into the culture medium was noted (Figure 5b), indicating cell death. Release of LDH into the culture medium is a commonly used method to determine cell lysis and cell death [49,50] and the absence or low amount of released LDH from the cells undoubtedly confirms that the reduced number of cells upon culture with the particles, especially the SA-MIPs, was not due to particles cytotoxicity.

Most of our current knowledge of how cells react to different kinds of perturbation is based on analysis of the response of an entire cell population. Microscopy techniques such as confocal microscopy, immunofluorescence microscopy, phase contrast microscopy and DHC are used to follow the behavior of living individual cells through time-lapse imaging. To scrutinize the effect of SA-MIPs on individual cells, DHC was used because of several advantages: low phototoxicity, label-free and quantitative data can be obtained on a single-cell level as well as on whole-cell populations [51]. Cell proliferation was studied for 48 h and showed a large reduction in the total number of RAW 264.6 cells in the presence of SA-MIPs, compared to cells grown in cell culture medium alone (Figure 6c). These results are well in agreement with the results obtained upon incubation of THP-1 cells with the examined particles and showed the effect of the particles on the whole population of the cells.

DHC also enables longitudinal tracking of individual cells to observe cell-to-cell variability within a population. This has previously been used to show that the JIMT-1 breast cancer cell line contains a subpopulation of cells with a decreased response to salinomycin compared to the rest of the population [25]. Longitudinal tracking of single RAW 264.7 cells incubated with SA-MIPs showed that most of the cells had an affected cell cycle in the presence of SA-MIPs (Figure 8a). These results demonstrate the presence of an augmented sub-population of almost non-dividing cells in the total cell population as well as normally dividing sub-populations upon culture in the presence of MIPs (Figure 8). Based on the results from flow cytometry and fluorescence microscopy that not all cells became associated with SA-MIPs, it can be assumed that the slowly or non-dividing cells were cells that in some way had been engaged with the SA-MIPs.

The average doubling time of the individually tracked cells can be estimated and was found to be roughly 14 h upon growth in a culture medium (Figure 8b). Although the variation in the estimated doubling time was relatively high, which was probably caused by using non-synchronized cells at the initiation of the experiments, these results fit well with an estimated doubling time of 20 h based on the growth curve of the entire population shown in Figure 6. An interesting observation is that proliferating cells in the presence of SA-MIPs, although of a minor number compared to cells grown in a culture medium, showed the same doubling time (Figure 8b).

Prolongation of the cell cycle due to cell–particle interactions that likely result in phagocytosis of the particles is an interesting observation. Based on the present results, the phagocytosing process overrules cell proliferation and when studying cell lines, the cell division and cell cycle were clearly affected (Figure 8). This is an interesting aspect, and it can be speculated if association with particles and thereby induction of the phagosomal pathway will generally re-program the cells. Wholesale re-programming will change the overall biological processes and pathways within all the cells in a cell line, and hence cell

proliferation will be arrested until the phagocytosed particles have been processed. A regulatory function of the phagosomes of macrophages and dendritic cells has previously been suggested [52] and the results presented here support this hypothesis.

## 5. Conclusions

We have demonstrated that SA-MIPs induced low induction and secretion of inflammatory cytokines that were comparable to phagocytosing cells cultured in the presence of the reference particles Alhydrogel and PLGA. The reduced number of cells upon incubation with SA-MIPs was not due to cytotoxicity but due to an attenuated cell cycle caused by the presence of the SA-MIP. The effects on the cell cycle were observed using a longitudinal investigation of individual cells and the results support the hypothesis that phagosomes also have regulatory functions with regard to the cell.

In summary, this suggests that synthesized MIPs would be applicable for future *in vivo* studies based on the low *in vitro* cytotoxicity and mild inflammatory properties of the SA-MIPs used in this study.

**Supplementary Materials:** The following supporting information can be downloaded at: https://www.mdpi.com/article/10.3390/app12126091/s1, Figure S1: Segmentation of a cell family; Figure S2: All cell family tress of tracked cells; Figure S3: SDS-PAGE analysis.

**Author Contributions:** Developed the concept, L.S., H.E., A.G.W. and K.R.; L.S. carried out the experiments and wrote the manuscript with support from K.A., A.G.W., K.G., K.R. and H.E.; synthesized the SA-MIPs, M.K. and K.G.; helped supervise the DHC experiments and data, K.A. and B.J.; conceived the study and were in charge of overall direction and planning, L.S. and H.E. All authors have read and agreed to the published version of the manuscript.

**Funding:** European Union's Horizon 2020 research and innovation program under the Marie Sklodowska-Curie grant agreement number: 721297. Biofilms Research Center for Biointerfaces and Malmö University, Sweden. Malmö University.

**Institutional Review Board Statement:** Not applicable.

**Informed Consent Statement:** Not applicable.

**Data Availability Statement:** The data underlying this article will be shared on reasonable request to the corresponding author.

**Acknowledgments:** The authors would like to thank Ravi Danielsson and Peter Falkman at the Department of Biomedical Science, Malmö University. The graphical abstract was created with Biorender.com.

**Conflicts of Interest:** The authors declare no conflict of interest.

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
