# Peer review of "Molecularly Imprinted Polymers Exhibit Low Cytotoxic and Inflammatory Properties in Macrophages In Vitro"

_applsci, doi:10.3390/app12126091_

Round 1
Reviewer 1 Report
The present study demonstrated the responses of THP-1 cells and RAW cells to the SA-MIPs treatment. It is an interesting study, however, the representative form of the data should be reconstructed. My comments are shown below:
1. The aim and conclusion of the present work were not directly given in the abstract, please add some concise statements.
2. In addition to the raw data shown in Fig1, I suggest adding a clear histogram to explain the flow cytometry result, what is the biological issue the result reflects? it should be clear to the readers.
3. Consistently, for Fig 1 and Fig 2, I suggest revising the figure titles to conclude "what the result reflects", rather than "what the results are".
4. No data is showing that the cell cycle was affected by the treatment. The division cannot directly reflect the cell cycle.
5. The cell number "2 x 104 cells per well" should be 2 x 104 cells, lines 206, 216.
Author Response
This is a letter from authors that concerns the revision of the study “Molecularly imprinted polymers exhibit low cytotoxic and inflammatory properties in macrophages in vitro”. Grammatic and linguistic corrections have been made. We are grateful that we have the possibility to give our reply to the reviewers and are happy to answer their questions. Below we have answered the questions in a point-by-point reply.
Comment and Suggestions from Reviewer 1
The present study demonstrated the responses of THP-1 cells and RAW cells to the SA-MIPs treatment. It is an interesting study, however, the representative form of the data should be reconstructed. My comments are shown below:
Point 1: The aim and conclusion of the present work were not directly given in the abstract, please add some concise statements.
Response 1:
The abstract has been given a clearer aim and conclusion, which is seen below.
“The aim of this study thus was to exemplarily assess whether SA-MIPs lead to inflammatory and/or cytotoxic responses when administered to phagocytosing cells in the innate immune system.” (page 1, line 16-18)
“In conclusion, SA-MIPs have a low degree of inflammatory properties, are not cytotoxic and can be applicable for future in vivo studies.” (page 1, line 27-29)
Point 2: In addition to the raw data shown in Fig1, I suggest adding a clear histogram to explain the flow cytometry result, what is the biological issue the result reflects? it should be clear to the readers.
Response 2:
The following text has been added to the Result section, page 6, clarifying the results shown by flow cytometry.
“The association is displayed in flow cytometry histograms showing the association of the NBD fluorophore-containing SA-MIPs (red histogram) or the lumogallion fluorophore-labeled Alhydrogel (blue histogram) with each cell line (Figure 1). The distribution of cells is seen on the x-axes with a magnification showing the background/autofluorescence of the cells in the absence of particles (black histogram Figure 1). Upon association with the fluorescent particles the cellular fluorescence will increase, shifting the histogram to higher FL1/FL2 values, and the percentage of cells associated with particles can be estimated. The results show that 100 % of the cells displayed an increased fluorescence upon incubation with Alhydrogel and hence are associated with the Alhydrogel particles (blue histogram Figure 1), whereas only 20-30 % of the cells became associated with the SA-MIPs (red histogram Figure 1).” (page 6, line 275-286)
Point 3: Consistently, for Fig 1 and Fig 2, I suggest revising the figure titles to conclude "what the result reflects", rather than "what the results are".
Response 3:
The figure title has been rewritten for both Figure 1 and Figure 2, see below.
“Figure 1. Particle association with phagocytosing cells.” (page 7, line 294)
“Figure 2. SA-MIPs association with RAW 264.7 cells.” (page 8, line 302)
Point 4: No data is showing that the cell cycle was affected by the treatment. The division cannot directly reflect the cell cycle.
Response 4:
The data show that SA-MIP treatment prolonged the first cell cycle after treatment with approximately 8 hours compared to the control (Figure 8). For the minority of cells that could complete further cell cycles, the cell cycle time for the second and third cell cycle was the same as for the control cells.
In Figure 8, each individual cell in 109 untreated cell families and 91 SA-MIPs treated cell families was tracked, making it possible to determine when each cell division took place and thereby also the length of each cell cycle. With this method we are not able to determine where in the cell cycle an effect is observed, but we clearly see that the first cell cycle is significant longer when the cells were treated with SA-MIPs compared to untreated cells.
Point 5: The cell number "2 x 104 cells per well" should be 2 x 104 cells, lines 206, 216.
Response 5:
Thank you for noticing. The cell number has now been changed to 2 x 104 cells, in both line 206 and 216.
Reviewer 2 Report
In this manuscript Louise Sternbæk, et al. described results of their work aimed at evaluation of effects of molecularly imprinted polymers against sialic acid (SA-MIPs) on the monocytic/macrophage cell lines to examine inflammatory properties and cytotoxicity of the particles. This field of investigation is significant because MIPs showed in in vitro models promising results for targeted interaction with different cell surface biomarkers, and these results indicate considerable potential of MIPS for in vivo applications.
For the study the authors used two cell lines for a number of in vitro models with application of modern adequate methods. The investigation revealed that SA-MIPs have low cytotoxicity and inflammatory properties, and the authors rightly assume that the results obtained suggest that these preparations are applicable for in vivo studies.
In general the study is well presented, proper controls are used and the conclusions are convincingly supported by experimental results, the data are of considerable novelty and interest. Manuscript is well written.
Several minor suggestions might improve the overall quality of the manuscript:
1. Page 2. Some conclusions based on the study results are described in the part “Introducnion” (line 83, 84, 86-90). It should be more correct to move this information in the part “Conclusion”.
2. Page 6, Table 1. There is a misprint “Alhydrogel, Single particle diameter, ~ 0.05 µm”, should be 0.5 µm as it is indicated in reference [30].
3. Page 6, line 266, 267. “Alhydrogel association with the cell lines (blue line) is much higher than seen for SA-MIPs”. The level of fluorescence is not a direct characteristic of association. The level of fluorescence depends in these experiments also on the quantum yield of the fluorophores.
In the results description the authors should also indicate that “20-30% of the cells became associated with SA-MIPs” as it is indicated in the part “Discussion”.
4. Page 9, Figure 4. The preparation demonstrated almost no dose dependence in experiments with analysis of TNF-α and IL-6 secretion. This fact should be discussed here, or in the part “Discussion”.
Author Response
This is a letter from authors that concerns the revision of the study “Molecularly imprinted polymers exhibit low cytotoxic and inflammatory properties in macrophages in vitro”. Grammatic and linguistic corrections have been made. We are grateful that we have the possibility to give our reply to the reviewers and are happy to answer their questions. Below we have answered the questions in a point-by-point reply.
Comment and Suggestions from Reviewer 2
In this manuscript Louise Sternbæk, et al. described results of their work aimed at evaluation of effects of molecularly imprinted polymers against sialic acid (SA-MIPs) on the monocytic/macrophage cell lines to examine inflammatory properties and cytotoxicity of the particles. This field of investigation is significant because MIPs showed in in vitro models promising results for targeted interaction with different cell surface biomarkers, and these results indicate considerable potential of MIPS for in vivo applications.
For the study the authors used two cell lines for a number of in vitro models with application of modern adequate methods. The investigation revealed that SA-MIPs have low cytotoxicity and inflammatory properties, and the authors rightly assume that the results obtained suggest that these preparations are applicable for in vivo studies.
In general the study is well presented, proper controls are used and the conclusions are convincingly supported by experimental results, the data are of considerable novelty and interest. Manuscript is well written.
Point 1: Page 2. Some conclusions based on the study results are described in the part “Introduction” (line 83, 84, 86-90). It should be more correct to move this information in the part “Conclusion”.
Response 1:
The last section in the introduction has been rewritten, to give a summary instead of the conclusion.
“In this study, we show that the induced release of inflammatory cytokines with SA-MIPs, was low in comparison with the reference particles Alhydrogel and PLGA. Moreover, the SA-MIPs were not cytotoxic to the cells, but the cell cycle was affected by the presence of the SA-MIPs as determined using DHC.” (page 3, line 90-95)
Point 2. Page 6, Table 1. There is a misprint “Alhydrogel, Single particle diameter, ~ 0.05 µm”, should be 0.5 µm as it is indicated in reference [30].
Response 2:
Reference 30 shows the particle size of Alhydrogel in 150 mM NaCl and in culture medium, however, Alhydrogel consists of nm sized particles (average 50 nm) that in PBS or 150 mM NaCl aggregate into particles with a size of 0.5 µm and higher. In the presence of proteins as serum, the aggregated particles further increase in size, resulting in particles with the size of several µm (Reference 31, Harris, J.R.; Soliakov, A.; Lewis, R.J. et al Micron, 2012, 43, 192-200 ).
Point 3: Page 6, line 266, 267. “Alhydrogel association with the cell lines (blue line) is much higher than seen for SA-MIPs”. The level of fluorescence is not a direct characteristic of association. The level of fluorescence depends in these experiments also on the quantum yield of the fluorophores.
In the results description the authors should also indicate that “20-30% of the cells became associated with SA-MIPs” as it is indicated in the part “Discussion”.
Response 3:
The text has been rewritten clarifying that an increased fluorescence compared to the cellular autofluorescence is used to determine if the cells are associated with the particles.
“The association is displayed in flow cytometry histograms showing the association of the NBD fluorophore-containing SA-MIPs (red histogram) or the lumogallion fluorophore-labeled Alhydrogel (blue histogram) with each cell line (Figure 1). The distribution of cells is seen on the x-axes with a magnification showing the background/autofluorescence of the cells in the absence of particles (black histogram Figure 1). Upon association with the fluorescent particles the cellular fluorescence will increase, shifting the histogram to higher FL1/FL2 values, and the percentage of cells associated with particles can be estimated. The results show that 100 % of the cells displayed an increased fluorescence upon incubation with Alhydrogel and hence are associated with the Alhydrogel particles (blue histogram Figure 1), whereas only 20-30 % of the cells became associated with the SA-MIPs (red histogram Figure 1).” (page 6, line 275-286)
Point 4: Page 9, Figure 4. The preparation demonstrated almost no dose dependence in experiments with analysis of TNF-α and IL-6 secretion. This fact should be discussed here, or in the part “Discussion”.
Response 4:
Text has been added to the Discussion section, see below.
“No distinct dose dependence of secreted TNF-α and IL-6 as a function of particle association was observed for either of the particles. In addition, SA-MIPs treatment did not cause an enhanced cytokine secretion compared to the reference particles, suggesting a limited NF-kappaB mediated inflammatory response was induced for the SA-MIPs.” (page 13, line 468-470)
Round 2
Reviewer 1 Report
The revised texts in the figure titles are not explicit. Please revise.
Author Response
This is a letter from authors that concerns the revision of the study “Molecularly imprinted polymers exhibit low cytotoxic and inflammatory properties in macrophages in vitro”. Grammatic and linguistic corrections have been made. We are grateful that we have the possibility to give our reply to the reviewers and are happy to answer their questions. Below we have answered the question in a point-by-point reply.
Comment and Suggestions from Reviewer 1
Point 1: The revised texts in the figure titles are not explicit. Please revise
Response 1: The figure titles for Figure 1 and Figure 2 has been revised, see below.
Figure 1. SA-MIPs had a lower association with phagocytosing cells compared to Alhydrogel.
Figure 2. Fluorescence microscopy images of the SA-MIPs’ potential intracellular location in RAW 264.7 cells.